evolution/biocomplexity/complexity

game theory, Prisoner's Dilemma game, chicken game, stag hunt game

**Author for correspondence:**
Jin Yoshimura
e-mail: yoshimura.jin@shizuoka.ac.jp

# A single 'weight-lifting' game covers all kinds of games

Tatsuki Yamamoto[1], Hiromu Ito[2,3], Momoka Nii[1], Takuya Okabe[1], Satoru Morita[1] and Jin Yoshimura[1,2,4,5]

[1]Graduate School of Integrated Science and Technology, Shizuoka University, Hamamatsu 432-8561, Japan
[2]Department of International Health, Institute of Tropical Medicine, Nagasaki University, Nagasaki 852-8523, Japan
[3]Department of Environmental Sciences, Zoology, University of Basel, Basel 4051, Switzerland
[4]Marine Biosystems Research Center, Chiba University, Uchiura, Kamogawa, Chiba 299-5502, Japan
[5]Department of Environmental and Forest Biology, State University of New York College of Environmental Science and Forestry, Syracuse, NY 13210, USA

HI, 0000-0001-9350-0546; TO, 0000-0001-7518-5837; SM, 0000-0001-5219-6218; JY, 0000-0003-1610-1386

Game theory has been studied extensively to answer why cooperation is promoted in human and animal societies. All games are classified into five games: the Prisoner's Dilemma, chicken game (including hawk–dove game), stag hunt game and two trivial games of either all cooperation or all defect, which are studied separately. Here, we propose a new game that covers all five game categories: the weight-lifting game. The player choose either to (1) carry a weight (cooperate: pay a cost) or (2) pretend to carry it (defect: pay no cost). The probability of success in carrying the weight depends on the number of cooperators, and the players either gain the success reward or pay the failure penalty. All five game categories appear in this game depending on the success probabilities for the number of cooperators. We prove that this game is exactly equivalent to the combination of all five games in terms of a pay-off matrix. This game thus provides a unified framework for studying all five types of games.

## 1. Introduction

Game theory was originally built as a theory on the optimization of individual economic behaviour against opponents [1], and it has been expanded to encompass evolutionary biology [2–4]. The most important issue in game theory is the dilemmas resulting from the discrepancy between the optimal strategy for an individual and that for the whole group [4–10]. These social dilemmas have been recognized frequently in daily life, e.g. dilemmas of vaccination [11] and dilemmas of traffic-lane change [12,13]. They are the primary factors that hinder the promotion of cooperation. Therefore, many studies have aimed

to explore the mechanisms that resolve the state of the dilemma conditions [4,6,7,14–16]. A canonical model used here is a two-person pairwise game with an unlimited well-mixed population, in which two players choose either cooperation (C) or defection (D) [14–16].

Two-person pairwise games are categorized into three types of dilemma games and two trivial games depending on the relative magnitudes of four elements in their $2 \times 2$ pay-off matrix: (1) Prisoner's Dilemma game (PD), (2) chicken game (CH; including hawk–dove game), (3) stag hunt game (SH), (4) all C (trivial C: TC) and (5) all D (trivial D: TD) [7,17].

In the PD game, known to have the strongest dilemma, it is very difficult to promote cooperation, as defection is the only strategy in its Nash equilibrium despite the best solution being the cooperation of all members [7,18]. It is also known as the two-player version of public goods games [19,20]. The CH game includes the famous hawk–dove game and snowdrift game [4]. In this game, the player receives greater damage when both defect than when he/she chooses cooperation but the opponent defects. This conflict often results in the stable coexistence of cooperators and defectors in a well-mixed population. The third type is the SH game, in which the player receives a greater benefit by choosing cooperation than defection when the opponent does cooperate [21]. Therefore, cooperation seems to be promoted more easily in the SH game than in the PD and CH games [7]. In the SH game, however, cooperation is likely to be disturbed because the benefit of both defectors is higher than that of an exploited cooperator. This situation in the SH game leads to two opposite Nash equilibria: all-defection and all-cooperation. Other than these three games with dilemmas, there are two trivial games with no dilemmas: all C (trivial C: TC) and all D (trivial D: TD).

Each of these five types of games is known to have a unique structure of dilemmas [7]. Because of this, these types of games are studied independently from each other. For example, the most difficult PD games have been studied extensively by introducing several reciprocity mechanisms [15]. In ecology, the hawk–dove game and other kinds of PD games have been studied to understand the evolution of cooperation in animals [2]. However, all three types of games (PD, CH and SH) and the two trivial games (TC and TD) could not be investigated in a single framework.

In this report, we propose a single game called a weight-lifting game that can evaluate all five categories together. In this game, a player has two choices: either carry (lift) a weight (cooperation: C) or pretend to carry it (defection: D); the possible combinations are (C, C), (C, D), (D, C) and (D, D). Both players gain a reward if a weight is successfully carried, but they suffer the penalty of failure if they fail to carry it. The success/failure of carrying the weight depends on the number of cooperators: the probability of success increases with the number of honest lifters.

This weight-lifting game becomes one of the five types depending on the specific values of the three probabilities $p_i$ ($i = 0, 1, 2$: the number of C). All five types of games are also quantitatively expressed by this game with a certain combination of success probabilities. Thus, the weight-lifting game is mathematically equivalent to all five categories of games. We also discuss the extension of this game to $N$-person games.

# 2. Model and results

In the weight-lifting game, baggage is carried by two players randomly selected from among an unlimited well-mixed population. Each player chooses a strategy from two choices: cooperation (C) in carrying the baggage by paying a cost ($c \geq 0$) or defection (D) without any cost. If the baggage is carried successfully, both players obtain a gain ($r > 0$) irrespective of his/her strategy. For example, the net gain of a player is $r - c$ if he/she cooperates in the successful case. By contrast, the gain is $r$ if he/she is defective but the baggage is successfully carried. In the unsuccessful case, both players pay a fine ($f > 0$) in addition to the cost $c$ of cooperation. Accordingly, the net gain is $-f - c$ and $-f$ for the cooperator and defector, respectively (figure 1a).

In terms of the probability of success $p_i$, where $i (=0,1,2)$ is the number of cooperators ($0 \leq p_i \leq 1$), we introduce two parameters: $\Delta p_1 = p_1 - p_0$ ($0 \leq \Delta p_1 \leq 1$) and $\Delta p_2 = p_2 - p_1$ ($0 \leq \Delta p_2 \leq 1$). These parameters express increments of the success probability by the presence of one cooperator. The difference between the success probabilities of two cooperators and no cooperator is $\Delta p_1 + \Delta p_2$ (figure 1b). The pay-off matrix of this game is represented in terms of the expected gains for two strategies (figure 1c):

$$F_C = rp_i - f(1 - p_i) - c \tag{2.1}$$

and

$$F_D = rp_i - f(1 - p_i). \tag{2.2}$$

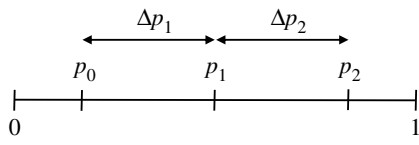

**(a)**

🧍: C    🧍: D

cost : c

fine : f

$1-p_0$    $p_0$    reward : r

$1-p_1$    $p_1$

$1-p_2$    $p_2$

failure    success

**(b)**

$\Delta p_1$    $\Delta p_2$

$p_0$    $p_1$    $p_2$

0    1

**(c)**

|   | C | D |
|---|---|---|
| C | $R : rp_2 - f(1-p_2) - c$ | $S : rp_1 - f(1-p_1) - c$ |
| D | $T : rp_1 - f(1-p_1)$ | $P : rp_0 - f(1-p_0)$ |

**(d)**

|   | C | D |
|---|---|---|
| C | $R : bp_2 - c$ | $S : bp_1 - c$ |
| D | $T : bp_1$ | $P : bp_0$ |

**Figure 1.** The weight-lifting game. (*a*) Two players lift the baggage (weight). A cooperator (C, white) pays a cost *c*, while a defector (D, black) does not. Each player either receives a reward *r* or pays a fine *f* depending on whether the lifting is successful. The success probability $p_i$ depends on the number of cooperators ($i = 0, 1, 2$). (*b*) We define $\Delta p_1$ and $\Delta p_2$ as the differences $p_1 - p_0$ and $p_2 - p_1$, respectively. Each of $\Delta p_1$, $\Delta p_2$ and $\Delta p_1 + \Delta p_2$ takes a numeric value between 0 and 1. (*c,d*) The pay-off matrix of the weight-lifting game. The matrix entries are collectively referred to by the letters R, S, T and P. (*c*) The pay-off matrix of the original game. (*d*) A simplified matrix of the same game, expressed in terms of $b = r + f$, the net benefit of success as measured relative to the failed case.

Note that $i = 2$ if the two players are both cooperative (C, C), $i = 1$ if one player is cooperative, (C, D) and (D, C), and $i = 0$ if neither player is cooperative (D, D). The pay-off matrix is simplified by introducing a parameter $b = r + f$ (>0), which is the benefit of a successful game compared to a failed game (figure 1*d*).

We consider a general set of C-D games satisfying three conditions $R \geq S$, $T \geq P$ and $T \geq S$ for the four matrix elements R, S, T and P (figure 2*a–e* black inequality symbols). The first two conditions, $R \geq S$ and $T \geq P$, express that cooperation increases pay-offs. The last condition, $T \geq S$, represents that players with different strategies benefit the defector more than the cooperator. The three conditions indicate $\Delta p_1 \geq 0$, $\Delta p_2 \geq 0$ and $c \geq 0$. In terms of the cost-to-benefit ratio $c/b$, the C-D games are classified according to whether the following inequalities are met (red inequality symbols in figure 2*a–e*).

(i) $S > P$, i.e. $\Delta p_1 > c/b$
(ii) $R > T$, i.e. $\Delta p_2 > c/b$
(iii) $R > P$, i.e. $\Delta p_1 + \Delta p_2 > c/b$.

Because $\Delta p_1$, $\Delta p_2 \geq 0$, the last inequality (iii) is automatically satisfied if either (i) or (ii) is satisfied. Accordingly, among four possibilities for whether or not conditions (i) and (ii) are met, only one (both met) is divided into two cases by the last condition (iii). In total, then, we have the following five cases to consider.

Trivial cooperation (TC): (figure 2*a*): $R > T \geq S > P$, where all three conditions (i), (ii) and (iii) are met. Both players are cooperative in the Nash equilibrium.

Chicken game (CH): (figure 2*b*): $T \geq R \geq S > P$, where (i) and (iii) are met. A different set of strategies, (C, D) and (D, C), is the Nash equilibrium.

Stag hunt (SH): (figure 2*c*): $R > T \geq P \geq S$, where (ii) and (iii) are met. The two strategies (C, C) and (D, D) are the Nash equilibrium.

Prisoner's Dilemma (PD): (figure 2*d*): $T \geq R > P \geq S$, where only (iii) is satisfied. The Nash equilibrium of (D, D) is not Pareto optimal.

Trivial defection (TD): (figure 2*e*): $T \geq P > R \geq S$, where none of the three conditions (i), (ii) and (iii) is met. Both players choose defection in the Nash equilibrium.

These results are summarized in figure 2*f*, which indicates that the game type is determined by the three parameters, $\Delta p_1$, $\Delta p_2$ and $c/b$. Every possible C-D game of all five types corresponds uniquely to a certain set of the three parameters (see electronic supplementary material).

# 3. Analysis of model

The five types of C-D games are shown in a three-dimensional parameter space ($\Delta p_1$, $\Delta p_2$, $b/c$) (figure 3*a*, *b*). Instead of $c/b$, the benefit-to-cost ratio $b/c$ is used as the third axis. While the former is convenient for

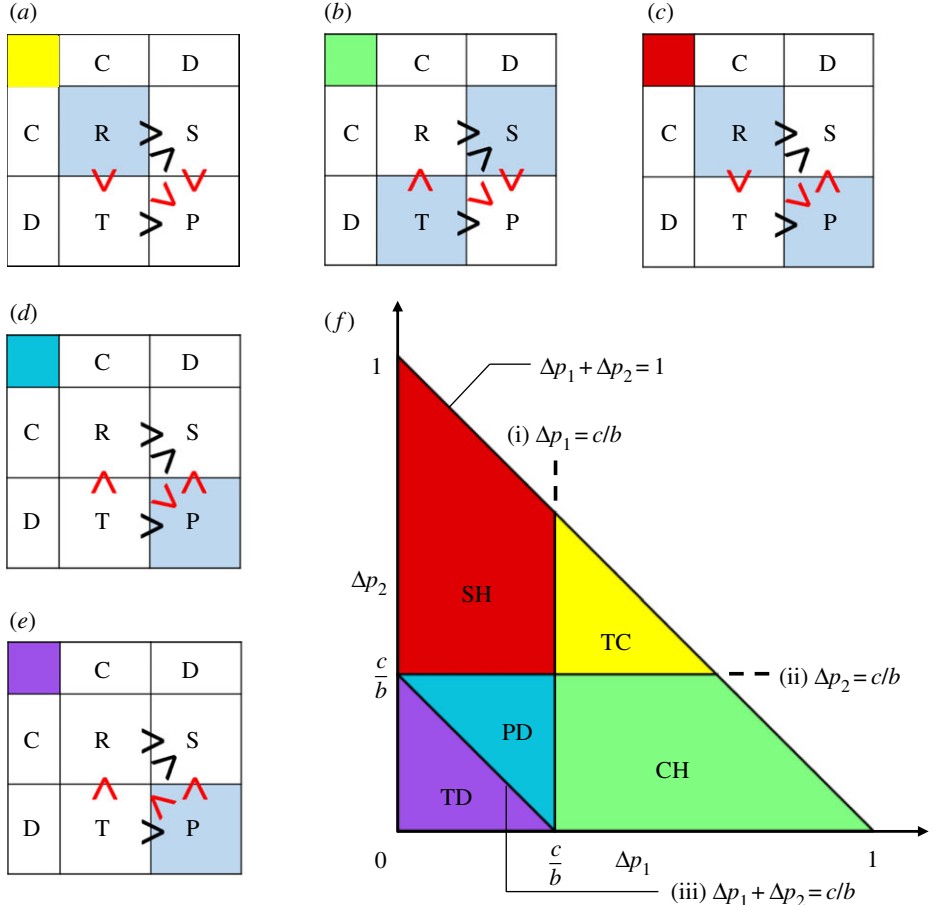

**Figure 2.** Five game types are classified according to the relative magnitudes of the pay-off matrix elements. (*a–e*) The relative magnitudes of four elements R, S, T and P are indicated by the sign of the inequality (>). Three black signs (>) are the preconditions for a general C-D game (i.e. cooperation increases the players' gain and defection has greater benefits than cooperation). There are five and only five cases (*a–e*), which are distinguished by the directions of three further inequalities (red): (*a*) trivial cooperation (TC), (*b*) chicken game (CH), (*c*) stag hunt game (SH), (*d*) Prisoner's Dilemma (PD) and (*e*) trivial defection (TD). (*f*) A phase diagram of the five game types is plotted on a $\Delta p_1 - \Delta p_2$ plane. Two parameters, $\Delta p_1$ and $\Delta p_2$, may take values within a rectangular triangle. Five regions for the five types are separated by three boundaries, which depend on the cost-to-benefit ratio $c/b$. Thus, all five games are obtained depending on the magnitude relationship of $\Delta p_i (i = 1, 2)$ and $c/b$.

use in mathematical expressions, the latter has a practical interpretation as an indicator of cooperation. The parameter regions for the five games are shown in colour: TC (yellow), CH (green), SH (red), PD (blue) and TD (violet). As the benefit-to-cost ratio $b/c$ increases, the volume in the parameter space of the cooperative game (TC) increases, while TD and PD, in which cooperation is not promoted, decline. For $b/c = 2$, TD (violet) and PD (blue) are dominant, while the TC game (yellow) is excluded (figure 3*a,c*). For $b/c = 5$, TC (yellow) becomes dominant, while TD (violet) and PD (blue) shrink (figure 3*b,d*). The ratio $b/c$ determines the three boundaries in a $\Delta p_1 - \Delta p_2$ cross-section (figure 3*c,d*). Cooperative behaviour (TC, SH) is promoted when $\Delta p_2$ exceeds the cost-to-benefit ratio $c/b$. The relative magnitudes of $\Delta p_1$ and $\Delta p_2$ determine the boundary between CH and SH, and the game type becomes SH for $\Delta p_2 > \Delta p_1$. This means that when the effect of an additional cooperator is increased synergistically, the number of cooperators increases. The CH game is obtained in the opposite case, $\Delta p_2 < \Delta p_1$.

Here we show the relationship between the game classes (TC, CH, SH, PD and TD) of the current model and the two dilemma strengths introduced recently in the studies of game theory [17,22,23]. The game classes are determined by two dilemma strengths: (1) gamble-intending dilemma (GID) and (2) risk-averting dilemma (RAD). The former (GID) is the strength of dilemma that players try to exploit each other, while the latter (RAD) is the strength of dilemma that players try to prevent from

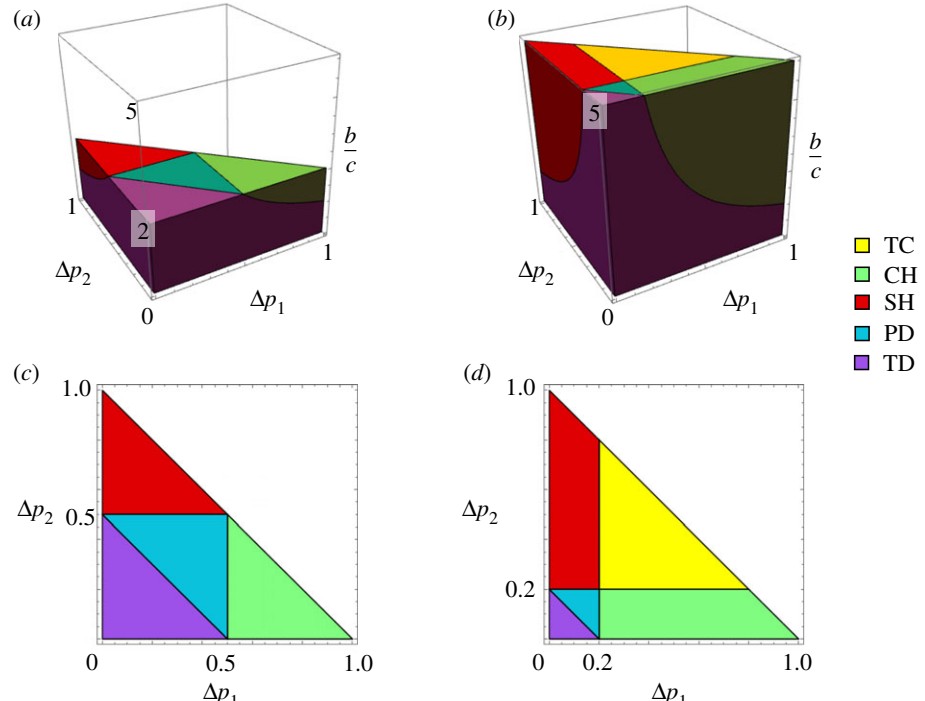

**Figure 3.** Phase diagrams of the weight-lifting game. (*a,b*) Diagrams in a three-dimensional space of ($\Delta p_1$, $\Delta p_2$, $b/c$). The third axis is the benefit-to-cost ratio $b/c$, the inverse of $c/b$. The domain of the three parameters forms a vertical triangular prism, which is divided into five regions for the five types of games. (*a*) $0 < b/c < 2$. (*b*) $0 < b/c < 5$. (*c,d*) Two-dimensional plots on a $\Delta p_1 - \Delta p_2$ plane, i.e. horizontal cross-sections of a three-dimensional diagram. (*c*) $b/c = 2$ ($c/b = 0.5$). (*d*) $b/c = 5$ ($c/b = 0.2$). The three boundaries separating the five regions are $\Delta p_1 = c/b$, $\Delta p_2 = c/b$ and $\Delta p_1 + \Delta p_2 = c/b$.

being exploited. They are given by the pay-off matrix elements (R, S, T and P in figure 1) as follows [17,22,23]:

$$\mathrm{GID} = \frac{\mathrm{T} - \mathrm{R}}{\mathrm{R} - \mathrm{P}}$$

and

$$\mathrm{RAD} = \frac{\mathrm{P} - \mathrm{S}}{\mathrm{R} - \mathrm{P}}.$$

Here the game class becomes TC when neither of the two dilemmas exists, because there is no dilemma situation. When only GID is positive, the game becomes CH. Conversely, when only RAD is positive, the game becomes SH. If both dilemmas are positive, the game becomes PD game. Note that RAD = 0 for $\Delta p_1 = c/b$ and GID = 0 for $\Delta p_2 = c/b$. As $\Delta p_1$ increases, RAD decreases while GID increases. On the other hand, as $\Delta p_2$ increases, GID decreases and RAD increases. As $c/b$ increases (from figure 4*d–f* to figure 4*a–c*), the region in which GID and RAD are both positive enlarges, while the TC region shrinks. Accordingly. Cooperative behaviour becomes difficult to achieve.

## 4. Discussion

We proposed a weight-lifting game that covers all five types of games. Previously, the resolution of dilemmas has been studied for each individual game (PD game, hawk–dove games, etc.). We can now analyse the dilemmas in a unified manner using this single model. Depending on the three parameters—$\Delta p_1$, $\Delta p_2$ and the cost-to-benefit ratio ($c/b$)—the game-theoretical behaviours of three dilemma games (CH, SH, PD) and two trivial games (TC, TD) are obtained in a unique manner. Therefore, players' optimal strategies may change as the parameters $\Delta p_1$, $\Delta p_2$ and $b/c$ vary. Because the change in game type represents a change in the strength of the dilemmas [7,22,23], it is an interesting future problem to investigate concrete mechanisms for varying the parameter values to promote cooperative behaviour by weakening the strengths of the dilemma (figure 4).

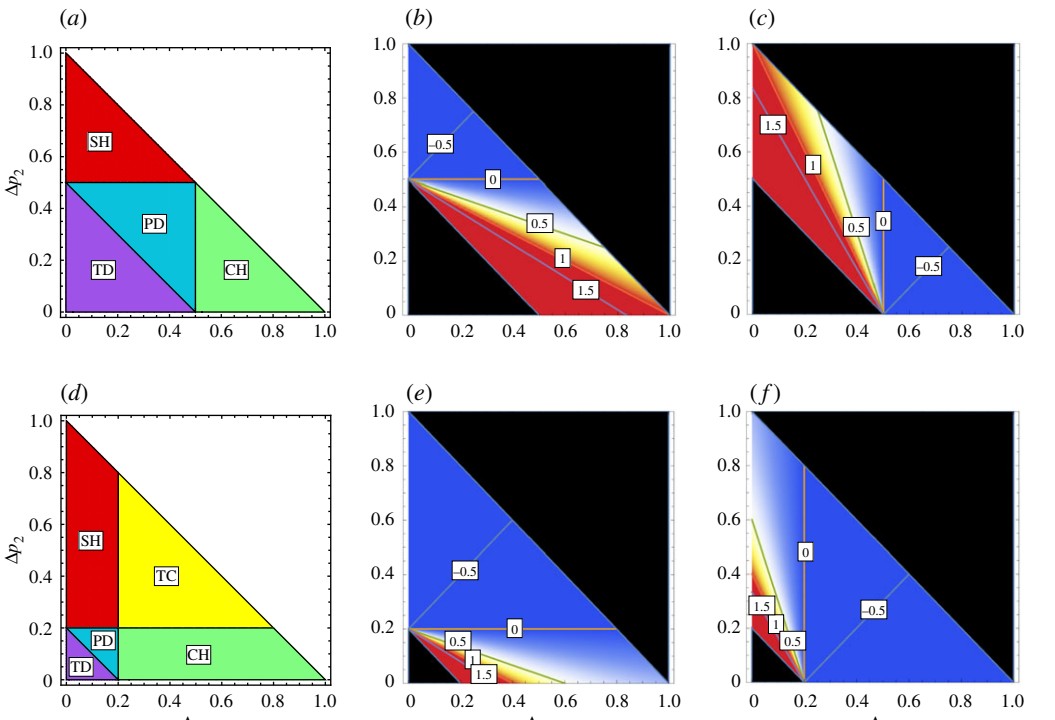

**Figure 4.** Relationship between game class and the dilemma strengths in the weight-lifting game. Two-dimensional ($\Delta p_1 - \Delta p_2$) phase plane with the condition of (*a–c*) $b/c = 2$ ($c/b = 0.5$) and (*d–f*) $b/c = 5$ ($c/b = 0.2$). (*a,d*) The colours indicate the regions of game class. (*b,e*) The strength of the gamble-intending dilemma (GID) and (*c,f*) the strength of the risk-averting dilemma (RAD) in SH, PD, CH and TC game [17,20,21]. The game class becomes TC when neither of the two dilemmas exists. When only GID is positive, the game becomes CH. Conversely, when only RAD is positive, the game becomes SH. If both dilemmas are positive, the game becomes PD game.

In the present model, the synergistic effect of cooperation [24] is represented in terms of $\Delta p_1$ and $\Delta p_2$, which are the increases in the success probability by a single cooperator when there is no or one other cooperator, respectively. The boundary between CH and SH games is determined by their relative magnitude.

It should be noted that CH and SH games address different dilemmas. The gamble-intending dilemma (GID) in the CH game is the dilemma in which players exploit each other. The risk-averting dilemma (RAD) in the SH game is the dilemma of avoiding exploitation by the opponent [7,17]. It is generally known that the promotion of cooperative behaviour is hampered more by GID than by RAD. In the PD game, which addresses both GID and RAD, the promotion of cooperation is hindered by these two dilemmas. The present model suggests that players' optimal behaviour may be changed by the synergetic effect [24]. Therefore, this model and its future developments should contribute to a full understanding of mechanisms for the promotion of cooperation.

Generalization to an *N*-player game appears to be a promising next step because it provides us with two advantages. The public goods game [25] has so far been focused on a single type of dilemma, e.g. the PD. The present game generalized to *N* players allows us to investigate all possible dilemmas in a unified manner. Another merit is that it allows investigation into how the synergetic effect works depending on the manner in which the success probability changes. A unified treatment of *N* players and the synergetic effect has not been made before, although the effect of *N* players and the synergetic effect [24] have been investigated separately.

In an *N*-player generalization of the present model, results different from that of the public goods game of a common type may be expected for the synergetic effect [25,26]. While the success probability $p_i$ increases as the number $i$ of cooperators increases, how the increment in $p_i$ varies determines whether synergy is effective or not. If $p_i$ increases linearly in $i$, the public goods game of a common type is obtained, in which cooperation is promoted when the expected utility $\Delta p \times (b/c)$ exceeds 1. By contrast, the success probability $p_i$ can be an arbitrary monotonic function in the present model. The synergetic effect depends on how $\Delta p_i$ varies. For example, in real-world society, it is expected that $p_i$ increases first slowly when there are a small number of cooperators, then moderately

as the number increases, and finally slowly again when the cooperators become a dominant majority. In this case, there is a threshold value for the number of cooperators to promote cooperation. Moreover, defection is promoted as the population is saturated with cooperators. It is an interesting future problem to develop these ideas as $N$-player game simulations of realistic scenarios.

The current model can be applied to the history of social revolution [27,28] to discuss the causal mechanisms of the revolution by analysing the dependence of success probability on the number of cooperators. In the early phase of a social system, many people adopt a cooperative attitude to enhance the probability of success. As society develops, the necessary number of cooperators decreases, and selfish people begin to proliferate. Accordingly, occupations not directly necessary for the survival of society increase as well. Thus, the upsurges of art and literature in ancient times as well as video games and entertainment in modern times may be considered to be the result of a surplus of resources in a developed society.

Data accessibility. This article has no additional data.

Author contributions. T.Y. and J.Y. conceived the study. T.Y. developed the model. All authors analysed the results and wrote the manuscript.

Competing interests. The authors declare no competing interests.

Funding. This work was partly supported by the Japan Society for the Promotion of Science (JSPS) KAKENHI (grant nos. 17J06741 and 17H04731 to H.I., grant no. 18K03453 to S.M., grant nos. 15H04420 and 26257405 to J.Y.).

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
