## [Reviewer comments · Royal Society Open Science]

Review History

RSOS-190799.R0 (Original submission)

Review form: Reviewer 1

Is the manuscript scientifically sound in its present form?

Yes

Are the interpretations and conclusions justified by the results?

Yes

Is the language acceptable?

Yes

Do you have any ethical concerns with this paper?

No

Have you any concerns about statistical analyses in this paper?

No

Recommendation?

Major revision is needed (please make suggestions in comments)

Comments to the Author(s)

This paper proposes a 'weight-lifting' game where group benefits will be inextricably related with individual choices. The cooperative dilemma situation proposed here is interesting, however, what is the difference between the model here and the public goods game in essence? Especially the continuous public goods game or with threshold. Essentially, successful tasks will benefit all the individuals, and failed task will lead to individual damage. Moreover, more cooperators will bring more contributions, and thus the task will be more likely accomplished. From this point of view, where is the innovation of this work? In the modeling section, the authors need to discuss or design their model more fully. Moreover, more adequate theoretical analysis is required.

Review form: Reviewer 2

Is the manuscript scientifically sound in its present form?

Yes

Are the interpretations and conclusions justified by the results?

Yes

Is the language acceptable?

Yes

Do you have any ethical concerns with this paper?

No

Have you any concerns about statistical analyses in this paper?

No

Recommendation?

Accept as is

Comments to the Author(s)

This work defined a new 2 by 2 game, called weight-lifting game, and was carefully analyzed from a theoretical standpoint. The concept the authors posed here seems quite interesting, and definitely new. Thus, I strongly suggest this MS should be published.

The game definition was deliberately in the body text and Fig. 1. It basically stands on the so-called Donor and Recipient (D & R) game, where $Dg' = c / (b-c) = Dr'$ is ensured, sub-class of PD. Because of the introduction of further game parameters; f , fine that is levied to both players when lifting & carrying is failed, and p_0 , p_1 , and p_2 ; success probability of lifting & carrying respectively when two defectors, one defector & one cooperator, and two cooperators are paring in a game, the observed game structure becomes more complex. On quite important issue, unlike the conventional 2 by 2 defined by payoff matrix; P , R , S and T , is that this particular game allows $R < P$. This is because D-dominant Trivial (they called TD) when Δp_1 ($:= p_1 - p_0$) and Δp_2 ($:= p_2 - p_1$) are both sufficiently small. As long as $R > P$ is ensured as in the conventional 2 by 2 games, the game becomes either PD, Chicken, SH or Trivial digressing from the conventional D & R game (that belongs to PD as I said).

All of those points are visually summarized in Fig. 3, which looks quite intelligible and impressive. It immediately teaches that the region of TD (violet) withers when the dilemma

strength becomes weak (b/c increasing). This entails the region of PD (blue) also shrinking and that of Trivial (yellow) swelling. It seems quite conservable.

So as to improve the MS be more impressive to the audience, let me suggest couple of things.

#1.

As I evaluated above, Fig.3 is quite nice. Yet it would be better, if heat-map of Dg' and Dr' only for PD, CH, SH and TC regions (i.e. except for TD region) would be presented aside panels (c) and (d).

#2.

Concerning the universal dilemma strength, the authors already cited two requisite literatures; one is the paper by Ito and another is by Wang. There is another work that gives the inception of the concept of Dg & Dr as below. It should be cited as well.

Tanimoto & Sagara; Relationship between dilemma occurrence and the existence of a weakly dominant strategy in a two-player symmetric game, *BioSystems* 90(1), 105-114, 2007.

Decision letter (RSOS-190799.R0)

27-Aug-2019

Dear Professor Yoshimura:

Manuscript ID RSOS-190799 entitled "A single 'weight-lifting' game covers all kinds of games" which you submitted to Royal Society Open Science, has been reviewed. The comments from reviewers are included at the bottom of this letter.

In view of the criticisms of the reviewers, the manuscript has been rejected in its current form. However, a new manuscript may be submitted which takes into consideration these comments.

Please note that resubmitting your manuscript does not guarantee eventual acceptance, and that your resubmission will be subject to peer review before a decision is made.

Your resubmitted manuscript should be submitted by 24-Feb-2020. If you are unable to submit by this date please contact the Editorial Office.

Kind regards,
Alice Power
Editorial Coordinator

on behalf of Professor Wen-Xu Wang (Associate Editor) and Miles Padgett (Subject Editor)
 openscience@royalsociety.org

Reviewers' Comments to Author:

Reviewer: 1

This paper proposes a 'weight-lifting' game where group benefits will be inextricably related with individual choices. The cooperative dilemma situation proposed here is interesting, however, what is the difference between the model here and the public goods game in essence? Especially the continuous public goods game or with threshold. Essentially, successful tasks will benefit all the individuals, and failed task will lead to individual damage. Moreover, more cooperators will bring more contributions, and thus the task will be more likely accomplished. From this point of view, where is the innovation of this work? In the modeling section, the authors need to discuss or design their model more fully. Moreover, more adequate theoretical analysis is required.

Reviewer: 2

Comments to the Author(s)

This work defined a new 2 by 2 game, called weight-lifting game, and was carefully analyzed from a theoretical standpoint. The concept the authors posed here seems quite interesting, and definitely new. Thus, I strongly suggest this MS should be published.

The game definition was deliberately in the body text and Fig. 1. It basically stands on the so-called Donor and Recipient (D & R) game, where $Dg' = c / (b-c) = Dr'$ is ensured, sub-class of PD. Because of the introduction of further game parameters; f , fine that is levied to both players when lifting & carrying is failed, and p_0 , p_1 , and p_2 ; success probability of lifting & carrying respectively when two defectors, one defector & one cooperator, and two cooperators are paring in a game, the observed game structure becomes more complex. On quite important issue, unlike the conventional 2 by 2 defined by payoff matrix; P , R , S and T , is that this particular game allows $R < P$. This is because D-dominant Trivial (they called TD) when $\Delta p_1 (= p_1 - p_0)$ and $\Delta p_2 (= p_2 - p_1)$ are both sufficiently small. As long as $R > P$ is ensured as in the conventional 2 by 2 games, the game becomes either PD, Chicken, SH or Trivial digressing from the conventional D & R game (that belongs to PD as I said).

All of those points are visually summarized in Fig. 3, which looks quite intelligible and impressive. It immediately teaches that the region of TD (violet) withers when the dilemma strength becomes weak (b/c increasing). This entails the region of PD (blue) also shrinking and that of Trivial (yellow) swelling. It seems quite conservable.

So as to improve the MS be more impressive to the audience, let me suggest couple of things.

#1.

As I evaluated above, Fig.3 is quite nice. Yet it would be better, if heat-map of Dg' and Dr' only for PD, CH, SH and TC regions (i.e. except for TD region) would be presented aside panels (c) and (d).

#2.

Concerning the universal dilemma strength, the authors already cited two requisite literatures; one is the paper by Ito and another is by Wang. There is another work that gives the inception of the concept of Dg & Dr as below. It should be cited as well.

Tanimoto & Sagara; Relationship between dilemma occurrence and the existence of a weakly dominant strategy in a two-player symmetric game, BioSystems 90(1), 105-114, 2007.

Author's Response to Decision Letter for (RSOS-190799.R0)

See Appendix A.

RSOS-191602.R0

Review form: Reviewer 1

Is the manuscript scientifically sound in its present form?

Yes

Are the interpretations and conclusions justified by the results?

Yes

Is the language acceptable?

Yes

Do you have any ethical concerns with this paper?

No

Have you any concerns about statistical analyses in this paper?

No

Recommendation?

Accept as is

Comments to the Author(s)

I think the authors has addressed all the comments of the reviewers. I support it to be published in this version.

Review form: Reviewer 2

Is the manuscript scientifically sound in its present form?

Yes

Are the interpretations and conclusions justified by the results?

Yes

Is the language acceptable?

Yes

Do you have any ethical concerns with this paper?

No

Have you any concerns about statistical analyses in this paper?

No

Recommendation?

Accept as is

Comments to the Author(s)

The revised MS, sufficiently improved, seems enough to be published as it is.

Decision letter (RSOS-191602.R0)

23-Oct-2019

Dear Professor Yoshimura,

I am pleased to inform you that your manuscript entitled "A single 'weight-lifting' game covers all kinds of games" is now accepted for publication in Royal Society Open Science.

You can expect to receive a proof of your article in the near future. To help expedite this process, we ask that you please send us the individual source file of each figure within your manuscript (high quality PDF or EPS files preferred) to:

Please contact the editorial office (openscience_proofs@royalsociety.org and openscience@royalsociety.org) to let us know if you are likely to be away from e-mail contact -- if you are going to be away, please nominate a co-author (if available) to manage the proofing process, and ensure they are copied into your email to the journal.

You have the opportunity to archive your accepted, unbranded manuscript, but access to the full text must be embargoed until publication.

Articles are normally press released. For this to be effective we set an embargo on news coverage corresponding to the publication date of the article. We request that news media and the authors do not publish stories ahead of this embargo (when final version of the article is available).

Kind regards,

on behalf of Professor Wen-Xu Wang (Associate Editor) and Miles Padgett (Subject Editor)
openscience@royalsociety.org

Reviewer comments to Author:

Reviewer: 1
Comments to the Author(s)

I think the authors has addressed all the comments of the reviewers. I support it to be published in this version.

Reviewer: 2
Comments to the Author(s)

The revised MS, sufficiently improved, seems enough to be published as it is.

Appendix A

Response to Reviewers

Reviewers' Comments to Author:

Reviewer: 1

This paper proposes a 'weight-lifting' game where group benefits will be inextricably related with individual choices. The cooperative dilemma situation proposed here is interesting, however, what is the difference between the model here and the public goods game in essence? Especially the continuous public goods game or with threshold. Essentially, successful tasks will benefit all the individuals, and failed task will lead to individual damage. Moreover, more cooperators will bring more contributions, and thus the task will be more likely accomplished. From this point of view, where is the innovation of this work? In the modeling section, the authors need to discuss or design their model more fully. Moreover, more adequate theoretical analysis is required.

RESPONSE:

We are bit surprised with this review. The reason is this. We showed with mathematical proof that the proposed game covers all kinds of pairwise games, not only Prisoner's dilemma game, but also chicken game (e.g., hawk dove game) and stag hunt game. The public goods game is known to be the multiplayer version of prisoner's dilemma game. It does not belong to chicken game, nor stag hunt game. This reviewer may not know this fact and believe that public goods game is the multiplayer version of all types of pairwise games. Unfortunately we were not careful enough to mention that the public goods game is just a particular case of our proposed game. We thought that it is well known that the PD games is a two-player version of public goods game, for example see below.

1. Fogarty TM. 1981. Prisoner's dilemmas and other public goods games. *Conflict Management and Peace Science* 5, 111-120.
2. Hauert C., Szabó G. 2003. Prisoner's dilemma and public goods games in different geometries: Compulsory versus voluntary interactions. *Complexity* 8, 31-38.
<https://doi.org/10.1002/cplx.10092>

We add a few sentences to indicate it explicitly and cite the above famous literatures in addition. See the text with blue background (lines 58, 200, 207-208, 280-284).

Reviewer: 2

Comments to the Author(s)

This work defined a new 2 by 2 game, called weight-lifting game, and was carefully

analyzed from a theoretical standpoint. The concept the authors posed here seems quite interesting, and definitely new. Thus, I strongly suggest this MS should be published.

The game definition was deliberately in the body text and Fig. 1. It basically stands on the so-called Donor and Recipient (D & R) game, where $Dg' = c / (b-c) = Dr'$ is ensured, sub-class of PD. Because of the introduction of further game parameters; f , fine that is levied to both players when lifting & carrying is failed, and p_0 , p_1 , and p_2 ; success probability of lifting & carrying respectively when two defectors, one defector & one cooperator, and two cooperators are paring in a game, the observed game structure becomes more complex. On quite important issue, unlike the conventional 2 by2 defined by payoff matrix; P, R, S and T, is that this particular game allows $R < P$. This is because D-dominant Trivial (they called TD) when Δp_1 ($:= p_1 - p_0$) and Δp_2 ($:= p_2 - p_1$) are both sufficiently small. As long as $R > P$ is ensured as in the conventional 2 by 2 games, the game becomes either PD, Chicken, SH or Trivial digressing from the conventional D & R game (that belongs to PD as I said).

All of those points are visually summarized in Fig. 3, which looks quite intelligible and impressive. It immediately teaches that the region of TD (violet) withers when the dilemma strength becomes weak (b/c increasing). This entails the region of PD (blue) also shrinking and that of Trivial (yellow) swelling. It seems quite conservable.

So as to improve the MS be more impressive to the audience, let me suggest couple of things.

#1.

As I evaluated above, Fig.3 is quite nice. Yet it would be better, if heat-map of Dg' and Dr' only for PD, CH, SH and TC regions (i.e. except for TD region) would be presented aside panels (c) and (d).

RESPONSE: Thank you for your suggestion. Due to the space in Figure 3, we added Figure 4 for the heat map of the two dilemma strength. This figure further clarifies the nature of the proposed game as a universal game. We add a paragraph explaining this new figure 4. We also modify the discussion to refer to the two dilemmas.

For changes, see the text with yellow background (lines 155-171, 180-183, .

#2.

Concerning the universal dilemma strength, the authors already cited two requisite literatures; one is the paper by Ito and another is by Wang. There is another work that gives the inception of the concept of Dg & Dr as below. It should be cited as well.

Tanimoto & Sagara; Relationship between dilemma occurrence and the existence of a weakly dominant strategy in a two-player symmetric game, BioSystems 90(1), 105-114, 2007.

RESPONSE: Thank you for your suggestion about the references. We added the above reference in the main text.

Journal Name: Royal Society Open Science

Journal Code: RSOS

Online ISSN: 2054-5703

Journal Admin Email: openscience@royalsociety.org

Journal Editor: Andrew Dunn

Journal Editor Email: openscience@royalsociety.org

MS Reference Number: RSOS-190799

Article Status: REJECTED

MS Dryad ID: RSOS-190799

MS Title: A single 'weight-lifting' game covers all kinds of games

MS Authors: Yamamoto, Tatsuki; Ito, Hiromu; Nii, Momoka; Okabe, Takuya; Morita, Satoru; Yoshimura, Jin

Contact Author: Jin Yoshimura

Contact Author Email: yoshimura.jin@shizuoka.ac.jp

Contact Author Address 1:

Contact Author Address 2:

Contact Author Address 3:

Contact Author City: Hamamatsu

Contact Author State:

Contact Author Country: Japan

Contact Author ZIP/Postal Code:

Keywords: game theory, hawk dove game, prisoner's dilemma, payoff matrix

Abstract: Game theory has been studied extensively in various fields of science. The major question addressed here is why cooperation is promoted in various societies, including human and animal societies. To solve this question, the prisoner's dilemma (PD) and hawk-dove game, which is a type of chicken game (CH), are often used in these studies. A canonical approach is to investigate the pairwise game, in which two players choose either cooperation (C) or defection (D). Based on the dilemma structure, all pairwise games are classified/categorized into five games: (1) three types of dilemma games, PD, CH and the stag hunt game (SH) and (2) two trivial games, all C (trivial C: TC) and all D (trivial D: TD). The problem of cooperation has been investigated in these games separately following the specific scenario of each. No single game, however, can cover all five categories. Here, we propose a new game that covers all five game categories: the weight-lifting game. The player choose either to (1) carry a weight (pay cooperation cost; C) or (2) pretend to carry it (pay no cost; D). The probability of success in carrying the weight depends on the number of cooperators, and the players either gain the success reward or pay the failure penalty. All five game categories appear in this game depending on the success probabilities for the number of cooperators. We prove that this game is exactly equivalent to the combination of all five games in terms of a payoff matrix. We also discuss its extension to N-person games.

EndDryadContent

27-Aug-2019 Tue, 27 Aug 2019 14:37:51

Dear Professor Yoshimura:

Manuscript ID RSOS-190799 entitled "A single 'weight-lifting' game covers all kinds of games" which you submitted to Royal Society Open Science, has been reviewed.